# Generative Art with Swarm Landscapes

**DOI:** 10.3390/e22111284

**Published:** 2020-11-12

**Authors:** Diogo de Andrade, Nuno Fachada, Carlos M. Fernandes, Agostinho C. Rosa

**Affiliations:** 1School of Communication, Arts and Information Technology, Lusófona University, 1749-024 Lisboa, Portugal; diogo.andrade@ulusofona.pt; 2HEI-Lab—Digital Human-Environment Interactions Lab, Lusófona University, 1749-024 Lisboa, Portugal; 3LARSyS: Laboratory for Robotics and Systems in Engineering and Science, Instituto Superior Técnico, University of Lisbon, 1049-001 Lisboa, Portugal; cfernandes@laseeb.org (C.M.F.); acrosa@laseeb.org (A.C.R.)

**Keywords:** particle swarm optimization, swarm art, function landscapes, procedural content generation, computational aesthetics, aesthetics model

## Abstract

We present a generative swarm art project that creates 3D animations by running a Particle Swarm Optimization algorithm over synthetic landscapes produced by an objective function. Different kinds of functions are explored, including mathematical expressions, Perlin noise-based terrain, and several image-based procedures. A method for displaying the particle swarm exploring the search space in aesthetically pleasing ways is described. Several experiments are detailed and analyzed and a number of interesting visual artifacts are highlighted.

## 1. Introduction

Particle Swarm Optimization (PSO) is a social intelligence model for learning and optimization that uses a population of particles to represent candidate solutions for a given problem [1,2]. The particle’s initial positions—i.e., the initial solutions—are typically randomized within the search space. PSO works iteratively, with particles moving through the problem landscape in search of better solutions. More specifically, at each step *t*, particles are informed of the best known position in the search space (or in the particle’s local network), using this information, together with the particle’s best known position, to guide how their velocity and position are updated, as shown in Equation (Equation 1) and Equation (Equation 2), respectively. When a particle’s position is updated, the solution it represents is evaluated according to a measure of fitness. This process is repeated until a “good enough” solution is found or a predefined maximum number of steps or evaluations is reached. The velocity and position of particle *i* for step *t* are updated according to [3]:(1)$$v_{i}\left(t\right)=\omega v_{i}\left(t-1\right)+c_{p}r_{p}\left(p_{i}^{\mathrm{best}}-x_{i}\left(t-1\right)\right)+c_{g}r_{g}\left(g_{i}^{\mathrm{best}}-x_{i}\left(t-1\right)\right)$$
(2)$$x_{i}\left(t\right)=x_{i}\left(t-1\right)+v_{i}\left(t\right)$$
where ω is the inertia weight parameter, which, together with the acceleration coefficients $c_{p}$ and $c_{g}$, allows the fine-tuning of the relative influence of the particle’s and the global (or local network) knowledge. The $r_{p}$ and $r_{g}$ parameters are random numbers taken from the uniform distribution in the interval [0,1] at each step *t*. Finally, $p_{i}^{\mathrm{best}}$ is the particle’s best known position, and $g_{i}^{\mathrm{best}}$ is the best position in the search space or in the particle’s local network. The second term in Equation (Equation 1) reflects the cognition aspect of PSO, i.e., a particle’s own thinking. The third term represents the social component, namely the collaboration between particles.

The value of the $g_{i}^{\mathrm{best}}$ parameter depends on the topology of the swarm, i.e., the group of particles with which any given particle *i* exchanges information. A global topology connects all particles to one another, while a local topology restricts communication of a particle to a subset of other particles. Common neighborhoods include a ring structure, in which particles are arranged in a ring, as well as 2D square lattices with von Neumann and Moore neighborhoods [2]. Topologies define the social interactions within a swarm and can have a considerable impact on its performance.

In this paper, we present a generative swarm art project capable of generating aesthetically engaging 3D animations by running a PSO algorithm over artificial landscapes. These landscapes are created by the same function being optimized by PSO, in the form f(x,y). For a given input (x,y), the function returns the height of the landscape and the fitness criterion at that point. Therefore, in the context of this paper, the velocity and position of a particle *i* are given by 2D vectors, i.e.,
(3)$$v_{i}=\left(v_{i,x},v_{i,y}\right)$$
(4)$$x_{i}=\left(x_{i},y_{i}\right)$$

Functions used for producing the artificial landscapes include well-known test functions for optimization, functions for procedural content generation (PCG), and images with pixel features such as brightness and saturation representing height/fitness. Particles move around in these landscapes in search of minima (and optionally, in the case of images, maxima), sometimes being trapped in local optima, offering interesting insights on the swarm’s social behavior. Furthermore, by manipulating parameters of the PSO algorithm, the generated visualizations allow for a deeper understanding of how PSO works in practice.

Landscape visualization can be configured in several ways. For example, landscapes can be rendered with height/fitness as a continuum along the visible electromagnetic spectrum. Height/fitness can be initially flat, and only rendered as the underlying function is explored by particles. Particle size is adjustable and particle communication lines can optionally be displayed. In summary, the swarm art project presented here provides a framework for visually exploring landscape-generating functions and a demonstration of how various PSO parametrizations self-organize in search of optimal points.

The paper is organized as follows: in Section 2, we review related work concerning the use of algorithms for generating artistic artifacts, with a special focus on swarm art. In Section 3, we describe the methods and software developed for creating the 3D animations, namely with respect to landscape aesthetics. Several experiments designed to highlight the proposed methodology are presented in Section 4; particular attention is given to their parameterization, as well as to interesting generated features. Section 5 includes a discussion on how these experiments were evaluated and selected for presentation. Section 6 closes the paper, offering some conclusions and suggesting alternative and/or complementary approaches.

## 2. Background

The term generative art refers to artistic objects that are partially or entirely created by autonomous systems, while computer-generated art, as defined by Boden and Edmonds [4], “results from some computer program being left to run by itself, with minimal or zero interference from a human being”. Accordingly, swarm art [5] can be defined as generative computer art that is created by artificial intelligence models with swarming behavior, requiring minimal or no interference from the artist. This is possible because swarm intelligence algorithms, which are basically populations of simple interacting entities that follow a set of rules to produce global patterns, have a considerable degree of autonomy and adaptability. Particle Swarm and Ant Colony Optimization [6] are examples of well-established swarm algorithms.

Swarm art is closely related to evolutionary art [7], but it has some specific characteristics that distinguish it from evolutionary and other types of generative or computer-generated art. One of the major differences between swarm and evolutionary art is that the former does not necessarily require an objective function. For swarm intelligence concur instead concepts like self-organization, emergency and stigmergy (indirect coordination through the environment) [8]. Their dynamic behavior and outcome are characterized by global complex patterns, rather than explicit goals imposed by objective functions. Swarm art systems have been used in several artistic conceptual spaces, like music [9] and multi-disciplinary installations [10], but they are best known for their applications in visual arts. In the remainder of this section, we will discuss visual swarm art that bears some relationship, either conceptual or aesthetic, with the proposed work.

Just like traditional painting, pictorial swarm art can be figurative and abstract. The swarm drawings by Semet et al. [11] are an example of a figurative approach. The authors use an artificial ant colony to make monochromatic drawings from black-and-white photographs. *Photogrowth* [12] is a similar approach, but in this case, the authors use artificial ants to produce non-photorealistic rendering of a given base-image, with a fitness function guiding image formation. Fernandes [13] used a swarm art system called *pherographia* to create artwork, which is in some aspects similar to those in [11,12].

Although figurative swarm art holds interesting aesthetic properties, abstract approaches can be more much exciting, both from conceptual and aesthetical points of view. From the conceptual perspective, it seems much more intuitive that swarms handle non-representational forms. The geometrical and life-like patterns generated by abstract-oriented swarms are more in harmony with their own properties. As for aesthetics, abstract styles allow for open-ended and informed exploration of search spaces, bringing novelty, surprise and an organic feel that somehow mirrors the essence of the medium. As seen in Section 4, the swarm visual forms presented in this work can be classified as abstract. The following are some examples of previous work on abstract swarm drawings and paintings.

Monmarché [14,15] was one of the first researchers to investigate the aesthetical possibilities of artificial ant colonies. Examples of Monmarché’s creations with swarms can be seen at the EA2013 Art&Science in Evolutionary Computation side-event catalogue [15]. Greenfield’s [16] first swarm art project also generated abstract images characterized by near-regular color patches. Later, Greenfield’s artwork evolved towards regularity and symmetry [17,18]. Urbano’s [19] ant paintings mimic circular sand wall building by the *Temnothorax albipennis* species. The images have an organic quality that is similar to other natural phenomena, like circular waves or microscopic organisms in fluids. Sand-bubbler crabs were the inspiration for Richter [20] to generate swarm paintings. The algorithm models the species collective behavior and generates colored images that recreate its feeding patterns and structures. Jacob et al. [21] proposed *SwarmArt*, an interactive swarm-based framework inspired by flocking birds. The interaction of the artist with the swarm is accomplished with video. However, the swarm is essentially autonomous and its dynamics can only be influenced to a certain degree. Finally, Fernandes et al. [22] used an ant clustering algorithm to design a swarm art system that was used to recreate famous abstract paintings.

## 3. Materials and Methods

### 3.1. Software

The swarm art project presented here is brought to life by the Visual PSO framework (https://github.com/DiogoDeAndrade/VisualPSO). Visual PSO was created with the Unity game engine [23], although its PSO capabilities are implemented in a separate library, OpenPSO.NET (https://github.com/fakenmc/openpso.net). This library is a C# reimplementation of OpenPSO, software originally written in C [2]. It includes a number of optimization benchmark functions, such as the Ackley [24], Griewank [25] or Rastrigin [26] functions, as well as a Perlin noise function [27], commonly used for PCG. The library allows users to easily create and use custom functions, a functionality used by Visual PSO, which adds several functions to handle images as an input for an optimization problem. Both tools are made available as open source software.

The application can be configured when running the executable binary on the command line. Running the application binary without any arguments will generate a random visualization. The available command line arguments are shown in Table 1. Alternatively, the source project can be opened in Unity and configured within the Unity editor.

### 3.2. Visualization

A function is visualized as a terrain, using a custom shader that displays the contour lines as glowing lines, as well as using a Fresnel-like effect to make the outlines of the search space more visible. The rendering also makes use of a post-process bloom effect to enhance the contoured nature of the generated landscapes and to create a more digital mood to the generated image.

Particles are represented as a sphere that leaves a trail, to make it easier to see the direction and speed of the particle. Particles are colored based on their current tendency: reddish while their fitness worsens, greener when it improves. Particle connectivity, i.e., the swarm topology, can optionally be shown by joining particles with a line, ignoring the *z*-buffer in order to avoid occlusion of the connection lines.

Visualizations can display the complete function topography from the start (the default), or only the explored search space, by activating the *fog of function* option. In this mode, the search space is initially displayed as flat terrain, and has the particles move, the underlying nearby terrain (i.e., fitness) is set to its correct height. This type of visualization allows the user to see where the particles have searched at any given moment. This is merely an aesthetic option, having no effect on how the PSO algorithm operates.

The camera movements are also generated procedurally. The camera can choose a random target or a particle (best, worst or random) to follow, and the movement itself is also generated considering the extents of the generated terrain. If the algorithm selects a particle to follow, a ray is cast onto the scene to check for occlusion; if an occlusion exists, a new position and target are selected. The camera also favors looking down into the scene, to avoid situations where it might be looking towards the sky. Every few seconds, the view changes to a new one.

### 3.3. PCG Landscapes

Most of our experiments focus on PCG landscapes. These are generated using an additive multi-layer Perlin noise [27] described by the following equation:(5)$$f\left(x,y\right)=\sum_{i=1}^{N}A^{i}P\left(f_{x}^{i}x+o_{x},f_{y}^{i}y+o_{y}\right)$$
where P(x,y) is the Perlin noise function, $A^{i}$ is the amplitude for layer *i*, $f_{x}^{i}$ and $f_{y}^{i}$ are the frequency for layer *i*, in the *x* and *y* axis, respectively; $o_{x}$ and $o_{y}$ are offsets for the terrain. On all our experiments involving the PCG terrain, $o_{x}=o_{y}=0$, $f_{x}^{i}=f_{y}^{i}$, $f_{x}^{i}=2^{i}f^{0}$ and $A^{i}={\frac{1}{2}}^{i}A^{0}$, where $f^{0}$ is the start frequency, and $A^{0}$ is the start amplitude.

There were alternative possible noise functions, such as Simplex [28] or Wavelet [29]. Perlin noise was chosen because of the low computational and implementation cost and lack of patents. Other techniques for terrain generation exist, such as erosion or fault-line algorithms [30], but most of these techniques require the whole terrain to be generated beforehand, which can be memory intensive and defeats the purpose of performing PSO.

### 3.4. Image Landscapes

Several image-based functions were also created, using the OpenPSO.NET custom function system. These functions accept an image and return a fitness value for a specific pixel. The first two functions are Saturation and Value. For these, the color at the desired pixel position is fetched, converted from RGB to HSV and then, either the Saturation or Value are used as fitness. Hue was discarded as a fitness function because in the case of minimization, for example, the swarm would just be searching for red or the closest color to red, which is not very interesting.

A function inspired on Ralph’s bell curve [31] was also designed and implemented. The premise is that many works of art consistently exhibit functions over color gradients that conform to a bell curve distribution. This allows parameters for a bell curve distribution to be computed to measure the response of a user to an image. Since a fitness value for a specific pixel is required, the concept of local response was introduced, in which the response is calculated in a customizable neighborhood of the pixel being evaluated instead of the whole image. The mean or the variance of the bell curve is then used as the fitness value. Although the PSO results are interesting, the function has a tendency to create spiky landscapes, which are not visually appealing when exaggerated for effect, and are too small to appreciate when not.

Currently, OpenPSO.NET only supports minimization problems, but in the case of images it is more interesting to search for local and global maxima. To circumvent this, the image functions can be inverted. Coupled with a negative scale in the landscape, the visual aspects do not change, but now the swarm is searching for the maximum.

## 4. Results

Our focus with this work was mainly in producing an appealing visualization for the algorithmic exploration. For this, PSO parameters were chosen that would lead to more interesting visualizations instead of those that would ensure faster convergence to a problem’s solution. As such, the particles’ maximum velocity and acceleration coefficients were limited, so that the viewer could follow the particles’ movements during the experiments. For all experiments, ω was set to 0.5, and, to ensure reproducibility, the random seed was set to 12345. The remaining parameters used on the different experiments are presented in Table 2.

### 4.1. Benchmark Functions

PSO algorithms are usually tested with benchmark functions such as the ones proposed by Ackley (Figure 1) or Rastrigin (Figure 2). Although these functions are typically too regular from a visual perspective, some of them are interesting for our goal of generating swarm art, as is the case of the Rastrigin function, used for **Experiment 1** (Figure 2, Appendix A). In this experiment, it is possible to observe that particles follow the function’s smooth yet pronounced curves towards convergence at the center position. Since this function is very regular, the choice of maximum velocity does not affect convergence quality too much, and as such, a very small one was used, enabling the viewer to appreciate the movement of the particles. A large scale factor for the particle representation was also employed, to draw the attention of the audience to the particle flow. For the color scheme, a purple/blue gradient was chosen, reinforcing the connection with the mathematical aspect of these functions with cold colors.

### 4.2. PCG Functions

The Perlin noise function, originally developed for PCG of natural-looking surfaces and textures, is at the core of this work, arguably creating more interesting landscapes, combining a smooth terrain with deep groves and mountains. Besides curiosity to see how a particle swarm would work over a procedurally generated landscape, these experiments were also designed to demonstrate how varying several of the PSO parameters influences the behaviors of the particles working on an easier-to-interpret landscape. All experiments in this section use a start amplitude, $A^{0}$, of 20, and a start frequency, $f^{0}$, set to 0.04.

In **Experiment 2** (Figure 3, Appendix A), an archetypical PCG landscape, created with eight layers of Perlin noise, is explored by a swarm of 49 particles. Using the fog of function option, it is clearly visible what areas the particles explore. In this experiment, a reasonable amount of the search space was visited. After a couple of seconds of simulation, the particles tend to reach local minima, although they do not stop searching for an improvement, balancing their own knowledge with the information provided by their Von Neumann neighborhood. For this experiment, the spectrum color palette was used, which maps the visible spectrum to the height of the function landscape, covering all the gamut of possibility.

In **Experiment 3** (Figure 4, Appendix A), the maximum speed of the particles was increased, which also enables more “risk taking”. The $c_{g}$ parameter was also increased, expanding the influence of the particle’s neighborhood on its behavior. As expected, there were less individual clusters, converging to one or two positions. The terrain is less detailed than in the previous experiment, with four layers of Perlin noise, becoming visible only when particles travel over it, due to the fog of function being enabled. This results in what seems as an island surrounded by high cliffs, all the more striking due to the direct association of color frequency with height.

For contrast, **Experiment 4** (Figure 5, Appendix A) maintains the level of detail (N=4), but switches the influences, increasing the pressure of the particle’s own experience versus the information provided by its topological neighbors. In this experiment, it can clearly be seen that there is less exploration of the space, since the particles tend to settle more in their own separate, individual regions, ignoring much of the information provided by their neighborhood. The color scheme was switched to a red/yellow gradient. With this aggressive palette, the ideas of conflict and isolation between the particles are reinforced. Regardless of color, the generated landscape is considerably different from the previous experiment, looking less like an island and more like a hole-ridden scene, with narrow, steep paths unfolding from its central area.

### 4.3. Image Functions

Another type of experiments created were based on a source image, from which a per-pixel fitness value is extracted. Two classical paintings (Figure 6) and different evaluation functions were used in our experiments.

In **Experiment 5** (Figure 7, Appendix A), the viewer can experience Juan Miró’s *Ciphers and Constellations in Love with a Woman* painting (Figure 6a) as an alien landscape. This extraterrestrial terrain fits well with Miró’s different perspective. The Value of the HSV representation of the color was used as the evaluation function. The Value was inverted—as well as the the scale of the terrain height—forcing the swarm to perform a maximization operation, instead of the default minimization. The reason for this is that the HSV color space is a cone and when Value is equal to zero, Saturation is irrelevant, so the particles would just search for the color black, while maximum brightness can be any color. The particle behavior in this experiment is not as interesting as in others, since searching for points of maximum brightness is not a difficult challenge in a bright painting such as this one.

In **Experiment 6**, Leonardo da Vinci’s *Mona Lisa* (Figure 6b) was selected as the source image. For the fitness value, the ideas behind Ralph’s bell curve were used, but, for aesthetic reasons, the stimulus instead of the response is used, computed in a local neighborhood with a radius of five pixels around the evaluated pixel. An interesting result of this experiment is that the particles tend to move towards the face of the Mona Lisa, as the stimulus value there was highest. Results from this experiment are shown in Figure 8 and Appendix A.

In both of these experiences, the original colors of the painting were used for the contour lines. Using one of the color schemes would make the fitness more obvious (Figure 9), but it would make the painting harder to recognize.

### 4.4. Connectivity Experiments

From an aesthetic perspective, the connectivity display is not the most appealing, but it provides compelling insights into the social component of PSO. For this, we performed four experiments (Figure 10) using the Perlin landscape function ($N=8,\;A^{0}=20,\;f^{0}=0.04$), changing only the $c_{p}$ and $c_{g}$ parameters, as well as the topology: Experiments 7, 8, and 9 use a 7×7 Von Neumann grid (total of 49 particles), while Experiment 10 uses a global topology with 25 particles. The green/blue color scheme was chosen on these experiments to provide contrast with the connection colors.

In **Experiment 7** (Figure 10a, Appendix A), $c_{p}$ is equal to $c_{g}$, i.e., the influence of the neighborhood knowledge and of the particles own experience is equally balanced. Besides the visual connections between particles, activated for this set of experiments, it is possible to observe a number of particle clusters forming around local minima, as well as a few isolated particles—some of which are perfectly static in deep, concentric minima. This experiment provides an instructive visualization, but its main raison d’etre is as a baseline for the next two experiments.

In **Experiment 8** (Figure 10b, Appendix A), the neighbors’ influence is five times that of the particles’ first-hand information. Looking at the generated visualization, this influence becomes quite clear when compared with the previous experiment: fewer and larger clusters form in valleys, with particles rarely seen wandering alone. The power of the crowds is obvious. Although this is an expected outcome, observing it as a swarm art process provides an excellent intuition of how PSO is working in general, and the influence of topology in particular.

Conversely, $c_{g}$ is set to zero in **Experiment 9** (Figure 10c, Appendix A). This means that the neighborhood has no influence at all on a particle’s decisions. It converts PSO into a multiple trials stochastic hill climber with a momentum term [30], where each particle acts on its own to find the optimum. Once again, the generated visualization confirms one’s reasoning, showing that particles essentially ignore each other, mostly hovering around minima near their spawn location.

**Experiment 10** (Figure 10d, Appendix A) is a very enlightening one. Parameters are the same as Experiment 7, except for the topology, which is now set to global, with a total of 25 particles. Thus, the whole swarm in considered as a particle’s neighborhood, which means that all particles are informed of the swarm’s best location at every step of the algorithm. A known issue with this approach is that it may lead the swarm to be trapped into local optima [32]. The intuition for this is strong: since all particles are biased towards the same point, the possibilities for exploration are reduced, and, with this, the chance of finding better locations. The visualization confirms our expectations, as a single large cluster of particles forms around a small region of the landscape; sometimes, a few particles wander off a bit, but never leaving the general area where the swarm is located.

## 5. Discussion

The swarm art project presented here is, in essence, a framework for exploring and experimenting with a parameterizable PSO algorithm working over landscape functions. The ten experiments presented in the previous section are primarily a showcase for what is possible to do with this framework. These experiments were selected by the authors based on their subjective aesthetical impact, similarly to [9,10,22], for example, and/or their didactic value, namely in how they highlight the influence of different PSO parameters on the optimization process. In this regard, the proposed VisualPSO framework can be viewed as an interactive art tool, where the main concern is not to objectively evaluate the aesthetics of a specific visualization, but instead to allow for user-controlled experiences [4] with a potentially educational facet.

## 6. Conclusions

In this paper, we have presented a method—and an associated software implementation—for generating swarm art in the form of expressive 3D animations. This was accomplished through the direct visualization of a parameterizable PSO algorithm running over a number of function-based artificial landscapes. The method was experimented on functions commonly used for benchmarking PSO variants, functions typically used for PCG and image-based functions. Besides the artistic side of the generated animations, the approach also presented an enlightening, didactic facet, where visualizations were able to confirm one’s intuition about the effects a given parameterization may have on the algorithm’s behavior. Taking this idea further, the generated swarm art could also potentially refute incorrect assumptions that the experimenter believes.

There are many potential avenues of research within or based on this work. For example, swarm algorithms such Ant Colony Optimization [6], Artificial Bee Colony Optimization [33] and others [34], could potentially produce visually appealing results of a different nature. Nonetheless, some care is required, since many of these bio-inspired swarm algorithms are in essence specific PSO setups [35]. Another interesting way of building upon this project would be to use some component of the PSO state to feed other generational components, such as the background or audio. A separate direction would be to create different types of aesthetics, rooted in the same concept; for example, instead of the landscape looking like a digital world, a more natural environment could be generated, creating photorealistic islands, and using particles to plant trees or carve rivers. One more possibility would be to add post-processing effects that give the whole visualization a completely different look.

A much different take would be to add an interactive component to the swarm landscapes, for example, by connecting the viewer to sensors and having these influence the function. This could be especially interesting by using a VR headset for full immersion, creating synergy between the art and the viewer, or even to expand this concept into games based on bio-feedback [36].

## Figures and Tables

**Figure 1 entropy-22-01284-f001:**
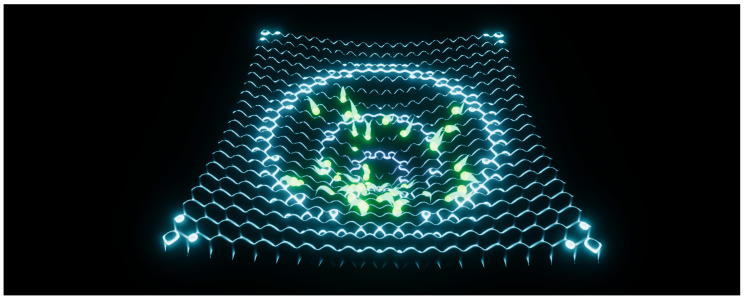
A visualization using the Ackley benchmark function.

**Figure 2 entropy-22-01284-f002:**
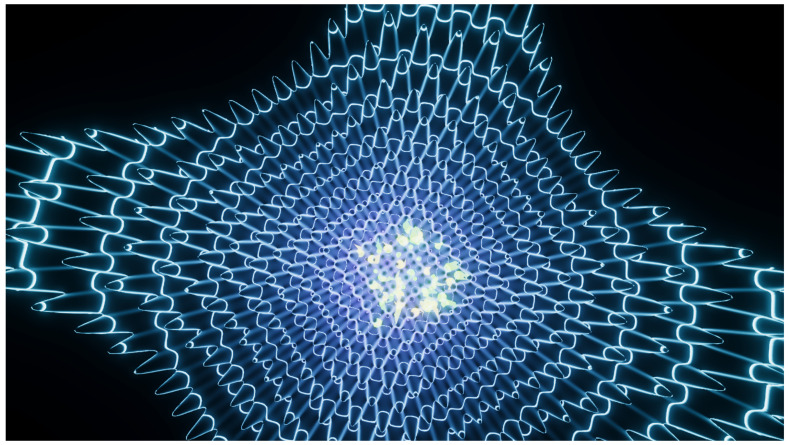
Experiment 1. Particles converging towards the center position of the Rastrigin benchmark function. See also Appendix A.

**Figure 3 entropy-22-01284-f003:**
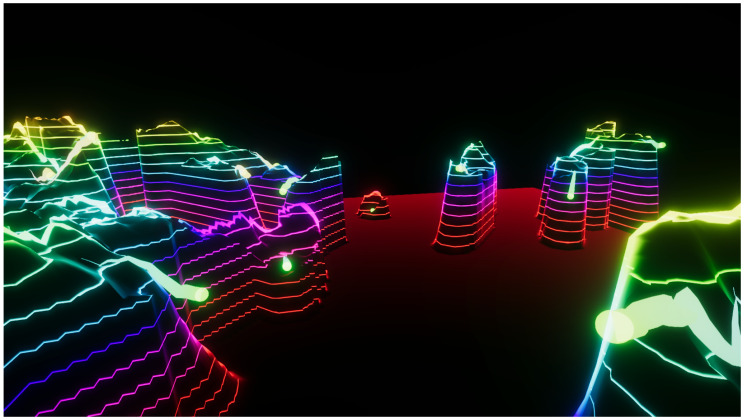
Experiment 2. Explored area of a Perlin landscape function ($N=8,\;A^{0}=20,\;f^{0}=0.04$). Regions with zero height were not explored by the particles. See also Appendix A.

**Figure 4 entropy-22-01284-f004:**
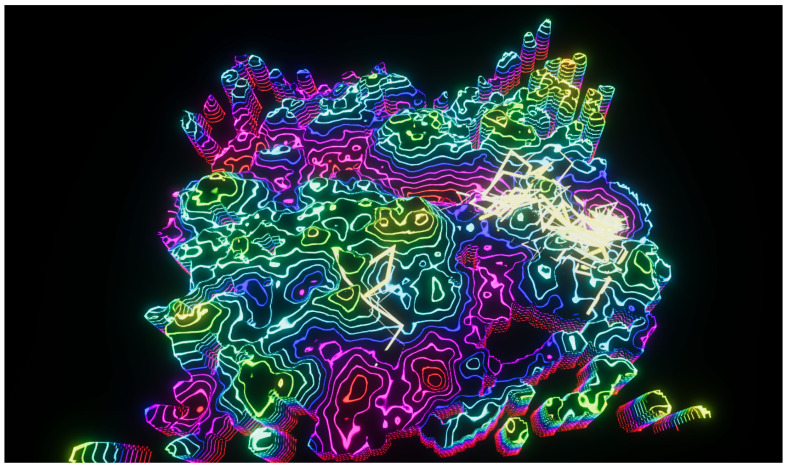
Experiment 3. Explored area of a Perlin landscape function. ($N=4,\;A^{0}=20,\;f^{0}=0.04$). Regions with zero height were not explored by the particles. In this experiment, the influence of the information provided by the particles’ topological neighbors is higher than that of the particle’s own experiences. See also Appendix A.

**Figure 5 entropy-22-01284-f005:**
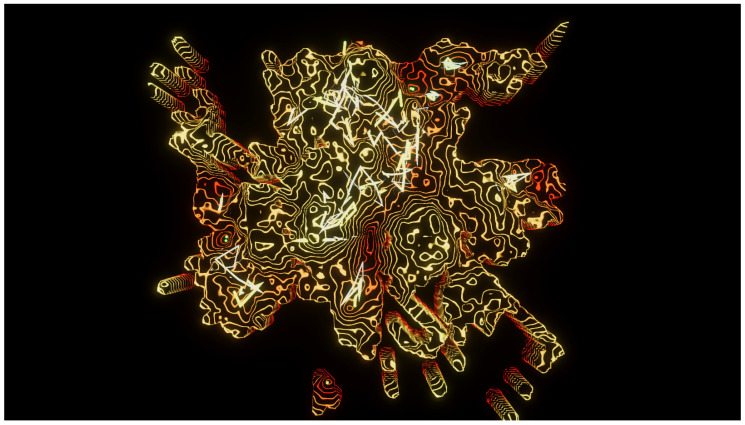
Experiment 4. Explored area of a Perlin landscape function ($N=4,\;A^{0}=20,\;f^{0}=0.04$). Regions with zero height were not explored by the particles. In this experience, the particles are highly individualistic, with low consideration for the information provided by their neighbors. See also Appendix A.

**Figure 6 entropy-22-01284-f006:**
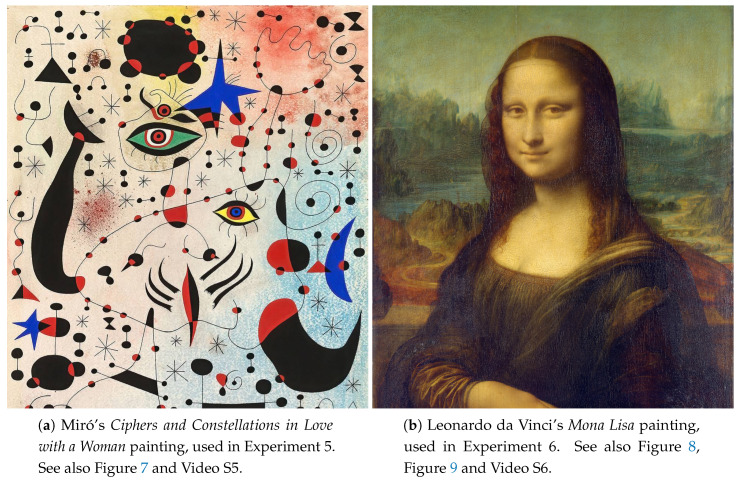
Paintings used in the image functions experiments.

**Figure 7 entropy-22-01284-f007:**
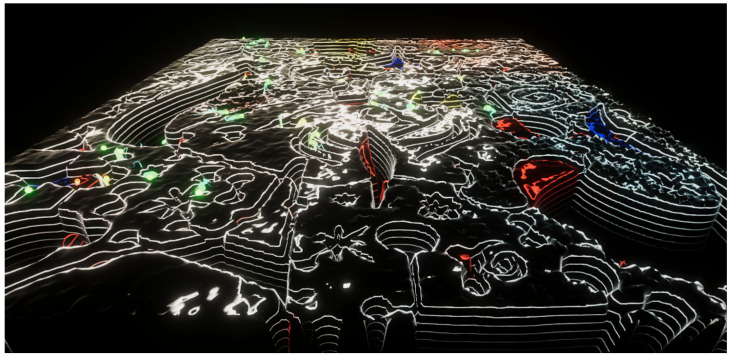
Experiment 5: Juan Miró’s *Ciphers and Constellations in Love with a Woman*. The fitness value is computed by converting the color to HSV space and taking the Value component (i.e., the brightness). See also Appendix A.

**Figure 8 entropy-22-01284-f008:**
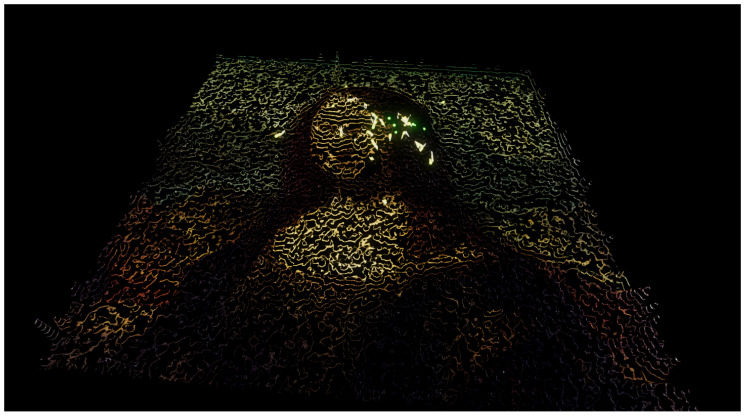
Experiment 6. Leonardo da Vinci’s *Mona Lisa*. The fitness value is measured by taking the mean of Ralph’s bell curve computed in the local neighborhood of the evaluated pixel. See also Appendix A.

**Figure 9 entropy-22-01284-f009:**
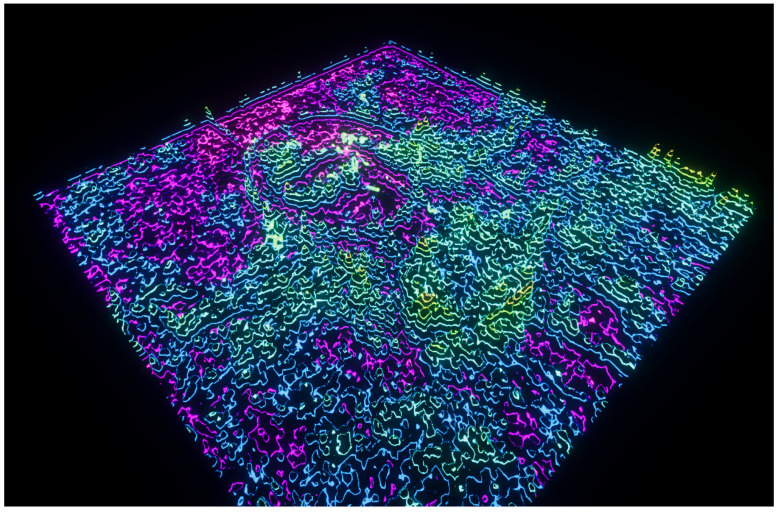
Alternative visualization of Experiment 6, where the color is derived from the fitness value on that point. The spectrum color palette was used in this case.

**Figure 10 entropy-22-01284-f010:**
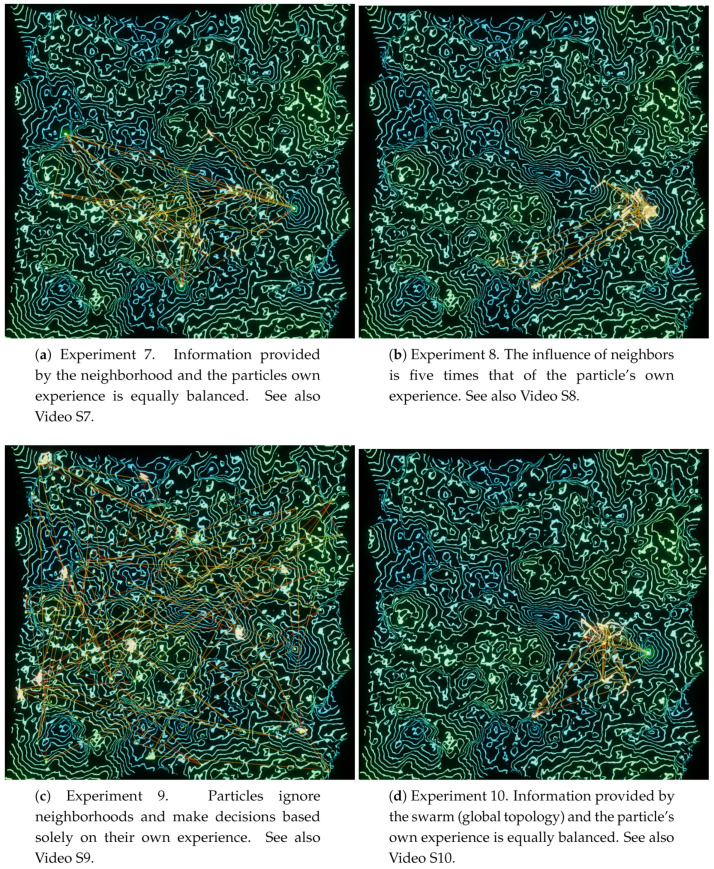
Connectivity experiments using Perlin noise ($N=8,\;A^{0}=20,\;f^{0}=0.04$) and a blue/green color scheme.

**Table 1 entropy-22-01284-t001:** Command line arguments for the VisualPSO application.

Argument	Description
-preset<number>	Select one of the builtin presets. Number must be an integer in the range [0,13].
-experiment<number>	Select one of the builtin experiments, as represented in this article. Number must be na integer in the range [1,10].
-random	Generate a random visualization. Same as not passing any parameter.
-rngseed	Seed for random number generator.
Perlin noise PCG options	
-landscape	Generate a Perlin-based function.
-octaves<number>	Number of octaves for the Perlin landscape.
-amplitude<value>	Initial amplitude for the Perlin landscape.
-frequency<value>	Initial frequency for the Perlin landscape.
Image options	
-imagesaturation	Use an image’s HSV Saturation as a function.
-imagevalue	Use an image’s HSV Value (brightness) as a function.
-ralphmean	Use Ralph’s bell curve mean as function.
-ralphvar	Use Ralph’s bell curve variance as function.
-sampleradius<radius>	Define the radius to use as the neighborhood for computing Ralph’s bell curve. Note that this is a $O(n^{2})$ operation, so large radius will take some time to compute.
-usestimulus	Use the stimulus value, instead of the more traditional response value for computing Ralph’s bell curve.
-useresponse	Use the response value for computing Ralph’s bell curve. This is the default.
-image<index>	Select a predefined image as the source image. Index must be na integer in the range [0,3].
-image<filename>	Load an image from the given path and uses it as source image. Only JPG and PNG are valid.
Benchmark functions	
-sphere	Use the sphere function.
-quadric	Use the quadric function.
-hyperellipsoid	Use the hyperellipsoid function.
-rastrigin	Use the Rastrigin function.
-griewank	Use the Griewank function.
-schaffer	Use the Schaffer function.
-ackley	Use the Ackley function.
-weierstrass	Use the Weierstrass function.
PSO parameters	
-w<value>	Specify ω.
-c<value>	Specify $c_{p}$ and $c_{g}$ with the same value.
-c1<value>	Specify the $c_{p}$ value.
-c2<value>	Specify the $c_{g}$ value.
-vmax<value>	Specify the maximum speed for the particles.
Visual parameters	
-scale<value>	Allow to scale the Y values of function.
-material<index>	Select the material to use for the visualization. Index must be an integer in the range [0,5].
-fof	Enable the fog of function option.
-connectivity	Display the particle connectivity.
-speed<number>	Define the speed of the simulation. Default is 1, 2 is twice the speed, 0.5 is half-speed.

**Table 2 entropy-22-01284-t002:** Parameters for the different experiments. “Exp.” is the experiment, “Scl.” is the scale, “PS” is the particle scale, “FoF” is the fog of function, “Conn.” indicates whether the connectivity/topology display is activated and “VN” means Von Neumann topology.

Exp.	Function	xMax	vMax	$c_{p}$	$c_{g}$	Topology, Swarm Size	Scl.	PS	FoF	Conn.	Material
1	Rastrigin	10	0.15	1.5	1.5	VN, 7×7	0.1	4	no	no	Purple/Blue
2	Perlin Lands.	100	0.5	1	1	VN, 7×7	1	1	yes	no	Spectrum
3	Perlin Lands.	100	10	1	4	VN, 7×7	1	1	yes	no	Spectrum
4	Perlin Lands.	100	0.5	2	0.5	VN, 5×5	1	1	yes	no	Red/Yellow
5	HSV Value	100	1	1	1	VN, 7×7	−20	1	no	no	Textured
6	Ralph’s Bell Curve Mean	100	1	1	1	VN, 7×7	−1	1	no	no	Textured
7	Perlin Lands.	100	5	1	1	VN, 7×7	1	1	no	yes	Blue/Green
8	Perlin Lands.	100	5	2	10	VN, 7×7	1	1	no	yes	Blue/Green
9	Perlin Lands.	100	5	5	0	VN, 7×7	1	1	no	yes	Blue/Green
10	Perlin Lands.	100	5	1	1	Global, 25	1	1	no	yes	Blue/Green

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
