# Peer review of "Generative Art with Swarm Landscapes"

_entropy, 2020, doi:10.3390/e22111284_

Round 1

Reviewer 1 Report

The authors present a modified particle swarm optimization (PSO) which generates swarm art. This is an instance of generative art producing 3D images and animations. The paper describes the design of the art making algorithm, discusses design parameters and show as visual examples some images and videos. I think this is a nice paper showing an application of a well-known bio-inspired computational paradigm, a PSO, for generating visual art. The examples shown are of aesthetic value and comparable with other recent works in the field. I recommend acceptance but would suggest to modify some details, which may enhance the clarity of the presentation.

  1. I think the title, Swarm Landscape, is a little misleading. The main topic of the paper is generating art. I feel this should be reflected in the title.
  2. The PSO is introduced by Eqs. (1) and (2). The coefficients $c_p$ and $c_g$ are generally known as cognitive and social weights. I recommend to introduce them accordingly, as this would also make it clearer that a PSO depends on both the cognitive abilities of each particle and its social context.
  3. I like the experiments with classical paintings where color properties are taken as a fitness to be fed into the PSO. From an aesthetic point of view I would find a comparison to the original painting helpful and also some experiments for making the resemblance more obvious.  

Author Response

  1. I think the title, Swarm Landscape, is a little misleading. The main topic of the paper is generating art. I feel this should be reflected in the title.

We thank the reviewer for this suggestion and have renamed the paper to “Generative Art with Swarm Landscapes”.

  1. The PSO is introduced by Eqs. (1) and (2). The coefficients $c_p$ and $c_g$ are generally known as cognitive and social weights. I recommend to introduce them accordingly, as this would also make it clearer that a PSO depends on both the cognitive abilities of each particle and its social context.

We agree with the reviewer and went one step further, highlighting the social context of the second and third terms of equation 1 in lines 26-27 of the manuscript, in a similar fashion to Shi and Eberhart’s original description.

  1. I like the experiments with classical paintings where color properties are taken as a fitness to be fed into the PSO. From an aesthetic point of view I would find a comparison to the original painting helpful and also some experiments for making the resemblance more obvious.  

Following the reviewer’s suggestion, the original paintings were added to the paper in Figure 6, allowing the reader to directly compare the paintings with the generated landscapes, while improving the flow of the text where this comparison is made (lines 234-251). In contrast, we found that adding more image experiments caused Section 4.3 to become somewhat large, repetitive and more difficult to follow, without adding much relevance to the presented results.

Reviewer 2 Report

This paper presents a method for performing generative art based on the concept of Particle Swarm Optimization. This manuscript is well written, provides the necessary background for understanding the topic, both in terms of generative art as well as with swarm optimization procedures. Besides, code is accessible through a public repository and examples of results are provided as images and video.

The paper itself merges two disciplines commented in the manuscript, the concept of self-generating art applied of 3D terrains with swarm optimization methods for their generation. For me the task is well addressed and well founded in terms of mathematical terms.

Nevertheless, I have some points I would like the authors to address regarding the actual evaluation of the results. Evaluation in terms of generative art is not a trivial issue since, in general, there is no clear objective to reach. Taking that into consideration, I am missing a proper evaluation based on actual metrics to actually check the performance of the proposal objectively. In case it is not possible to provide such metrics due to the nature of the task, please provide some literature background supporting the fact that this type of tasks are not evaluated in those terms.

Finally, I am missing a comparative work to other alternative algorithms tackling this task. I would suggest the authors perform such comparison and, if not, provide the proper arguments not to perform it.

Author Response

Evaluation in terms of generative art is not a trivial issue since, in general, there is no clear objective to reach. Taking that into consideration, I am missing a proper evaluation based on actual metrics to actually check the performance of the proposal objectively. In case it is not possible to provide such metrics due to the nature of the task, please provide some literature background supporting the fact that this type of tasks are not evaluated in those terms.

We thank the reviewer for this valuable suggestion and have added a new section (Section 5 – Discussion), where we discuss the criteria used for selecting the ten experiments presented in the previous section.

Finally, I am missing a comparative work to other alternative algorithms tackling this task. I would suggest the authors perform such comparison and, if not, provide the proper arguments not to perform it.

This is an interesting issue, and the answer is multifold, depending on how we interpret it:

  • To the best of our knowledge, swarm algorithms have not been used for tackling the exact task performed by the VisualPSO framework, so a comparison with previous work for this exact task would be difficult to address. Previous use of swarm algorithms for generating art in general, as well as the distinction between swarm and evolutionary algorithms in this regard, is presented in Section 2 – Background.

  • If by “task” the reviewer means optimization in general, then yes, there are countless other algorithms. The exact task performed by VisualPSO was set up in a way that mainly swarm algorithms make sense – i.e., having a “swarm” of particles searching the solution space in a continuous fashion. Using an evolutionary algorithm, for example, would likely produce less interesting visualizations, given that individuals in new generations typically replace discarded individuals, thus making solutions “jump” to different parts of the landscape instead of continuously moving through it between iterations. As such, a comparison with other approaches would make sense essentially within the context of swarm algorithms.

  • Since swarm algorithms have not been used for this exact task (to the best of our knowledge), a comparison would require that we implemented the alternative algorithms and tested them ourselves. Within the multitude of existing swarm algorithms (many of which described in [1], for example), there are some such as Ant Colony Optimization and Artificial Bee Colony Optimization that would certainly be interesting to observe. There are two issues with implementing and testing alternative algorithms, however:

    1. For each added algorithm actually different from PSO (see next point), the paper would dramatically increase in size, requiring a specific introduction to the algorithm, additional tables and descriptions of the particular algorithm’s parameters, a number of new experiments and description of the associated results, etc. In short, we believe that experimenting with a different swarm algorithm is worthy of a new paper altogether. Even within the context of PSO, we are only tackling a small subset of common parametrizations and setups [2] just to show some of the possibilities of the proposed approach.

    2. Many swarm algorithms are essentially subsets of PSO [3], and as such, it is debatable whether these would produce results substantially different from the ones presented in this paper. In any case, and taking into account the issue raised by the reviewer, we added a note to the conclusions (lines 305–308), highlighting that a possible research path following this work is to experiment with different swarm algorithms while taking care to select methods that are actually different from PSO.

[1] Del Ser, J.; Osaba, E.; Molina, D.; Yang, X.S.; Salcedo-Sanz, S.; Camacho, D.; Das, S.; Suganthan, P.N.; Coello, C.A.C.; Herrera, F. Bio-inspired computation: Where we stand and what’s next. Swarm and Evolutionary Computation 2019, 48, 220–250. doi:10.1016/j.swevo.2019.04.008.

[2] Sengupta, S., Basak, S., & Peters, R. A. (2019). Particle Swarm Optimization: A survey of historical and recent developments with hybridization perspectives. Machine Learning and Knowledge Extraction, 1(1), 157-191.

[3] Villalón, C.L.C.; Stützle, T.; Dorigo, M. Grey Wolf, Firefly and Bat Algorithms: Three Widespread Algorithms that Do Not Contain Any Novelty. Swarm Intelligence, 12th International Conference, ANTS 2020; Dorigo, M.; Stützle, T.; Blesa, M.J.; Blum, C.; Hamann, H.; Heinrich, M.K.; Strobel, V., Eds. Springer, 2020, pp. 121–133.